# Optimization of Tokuhashi Scoring System to Improve Survival Prediction in Patients with Spinal Metastases

**DOI:** 10.3390/jcm11185391

**Published:** 2022-09-14

**Authors:** Hung-Kuan Yen, Chih-Wei Chen, Wei-Hsin Lin, Zhong-Yu Wang, Chuan-Ching Huang, Hsuan-Yu Chen, Shu-Hua Yang, Ming-Hsiao Hu

**Affiliations:** 1Department of Orthopaedic Surgery, National Taiwan University Hospital, Taipei 100, Taiwan; 2School of Medicine, National Taiwan University College of Medicine, Taipei 100, Taiwan; 3Department of Orthopaedic Surgery, National Taiwan University Hospital, Hsin-Chu Branch, Hsinchu 300, Taiwan; 4Department of Orthopaedic Surgery, National Taiwan University College of Medicine, Taipei 100, Taiwan

**Keywords:** body mass index, neoplasm staging, spine metastases, surgical oncology, survival, target therapy

## Abstract

Introduction: Predicting survival time for patients with spinal metastases is important in treatment choice. Generally speaking, six months is a landmark cutoff point. Revised Tokuhashi score (RTS), the most widely used scoring system, lost its accuracy in predicting 6-month survival, gradually. Therefore, a more precise scoring system is urgently needed. Objective: The aim of this study is to create a new scoring system with a higher accuracy in predicting 6-month survival based on the previously used RTS. Methods: Data of 171 patients were examined to determine factors that affect prognosis (reference group), and the remaining (validation group) were examined to validate the reliability of a new score, adjusted Tokuhashi score (ATS). We compared their discriminatory abilities of the prediction models using area under receiver operating characteristic curve (AUC). Results: Target therapy and the Z score of BMI (Z-BMI), which adjusted to the patients’ sex and age, were additional independent prognostic factors. Patients with target therapy use are awarded 4 points. The Z score of BMI could be added directly to yield ATS. The AUCs were 0.760 for ATS and 0.636 for RTS in the validation group. Conclusion: Appropriate target therapy use can prolong patients’ survival. Z-BMI which might reflect nutritional status is another important influencing factor. With the optimization, surgeons could choose a more individualized treatment for patients.

## 1. Introduction

Bone metastases are mostly located in the spine (>50%), frequently involving thoracic vertebrae (60–70%), followed by lumbar vertebrae (15–30%) and rarely cervical vertebrae (<10%) [1]. Approximately half of the patients with metastatic spine tumor have lesions at multiple levels [2]. For all spinal metastases, the most common primary malignancy sites are breast, lung, prostate, and kidney [3,4,5]. Spinal metastases can cause acute symptoms, such as neurological impairment and intolerable pain, which suggests an advanced stage of metastatic disease and even significant morbidity. Despite advances in radiotherapy, chemotherapy, target therapy, and hormone therapy, treating spine metastases is challenging for spinal surgeons. One therapy should be chosen to provide the maximum palliative effect and to improve the quality of life of patients with minimum operative morbidity and mortality [6,7,8]. Survival is a significant determinant of deciding treatment plan among other important factors. Orthopedic and neurosurgeons could consider excisional procedures, including extensive curettage or en bloc resection of the vertebral body for patients with good prognosis since undertreatment might lead to tumor recurrence, further decreased quality of life, or even revision surgery. In contrast, more conservative treatment should be considered for patients with shorter life expectancy since they might not have enough time to recover and benefit from the surgery.

To facilitate decision making, Tokuhashi et al. published a scoring system in 1990 [9] and revised it in 2005 [10]; this scoring system is most widely used for predicting survival time in patients with spine metastases [11]. The authors individuated six parameters for analysis, namely general condition, extraspinal bone metastases, number of metastases in the vertebral body, visceral metastases, primary site of metastases, and neurologic deficit severity. Except for the primary site, a score of 0–2 is assigned for each parameter. The primary tumor domain is awarded a score of 0–5, making the maximum score 15, which indicates the best prognosis. Patients with scores of ≤8 are predicted to survive <6 months, and those with scores of ≥12 are predicted with a survival ≥12 months. However, with the progress of all types of therapy and imaging techniques, patients with spinal metastases live longer than before, and the accuracy of revised Tokuhashi score (RTS) has been declining with time [11,12,13,14,15]. One previous study specifically stated that the prediction is less accurate for patients with an estimated survival of less than 6 months [12]. Hence, in this study, we intended to create a more predictive model by adding new variables to the existing RTS in order to distinguish the patients with a survival ≥6 months from those with <6 months [14,15].

## 2. Materials and Methods

### 2.1. Patients

The retrospective research was approved by the institutional review board of the hospital (approval no.: 202005016RINC). We enrolled patients who underwent surgical intervention for pathologically confirmed spinal metastasis, including palliative decompression and aggressive excisional surgery, between January 2012 and December 2017, in a tertiary center in Taiwan. We eliminated 136 cases that had more than 3 months in between date of CT scanning and their surgery because this might lead to inaccurate metastases evaluation. Four pediatric patients were excluded because of differences in their cancer characteristics, significantly good prognosis [16,17]. Five patients were excluded since their 6-month survival could not be ascertained (Figure 1). In total, 171 patients who underwent surgery before 31 December 2016, referred to as the reference group, were used to evaluate the accuracy of RTS and determine factors that affect prognosis. The remaining 57 patients who underwent spinal surgery after 1 January 2017, referred to as the validation group, were used to validate the reliability of a new score, that is, adjusted Tokuhashi score (ATS; Figure 1).

### 2.2. Medical Information

We recorded the medical information of each patient, including age, sex, survival time (calculated from the time of surgery until death), primary tumor type, height, weight, body mass index (BMI), RTS, and adjuvant therapy type (e.g., chemotherapy, target therapy, and hormone therapy). The primary tumor type was determined according to pathological reports. If the pathological reports indicated unknown origin or unspecified tissue type, previous medical records were referenced to speculate its primary site. Mean and standard deviation of BMI of the general population were accessed through Ministry of Health and Welfare in Taiwan to transform the patients’ BMI into the corresponding Z-scores (Z-BMI) [18]. Death was defined as the time between the patient ’s first surgery for a spinal metastasis and death of any cause. To avoid immortal time bias, all data were censored at patients’ death or two years after the index surgery.

### 2.3. Blinding

All medical details were recorded by two independent individuals. The one who recorded factors such as the number of vertebrae involved and the resectability of visceral metastasis was blinded for the patient’s prognosis. A third author performed all the statistical analysis.

### 2.4. Statistical Analysis

The data are presented as means ± standard deviations, and the percentages and ranges are presented in parentheses. The comparisons of continuous variables between different groups were analyzed using one-way analysis of variance. Discrete variables were analyzed using the chi-square test, and Yates correction was applied if the expected number of patients was low. Logistic regression was used to determine reliable factors for predicting the 6-month survival rate. Survival curve and its *p* value were obtained through the Kaplan–Meier method and log-rank test. Receiver operating characteristic (ROC) curve and area under curve (AUC) was applied to compare ATS and RTS. A *p*-value of <0.05 was considered significant, and R (version 4.0.2) was used for statistical analysis.

## 3. Results

### 3.1. Reference Group

Of the 171 patients, 103 (59%) were male and 68 (41%) were female. Mean age at the time of surgery was 56.0 ± 9.9 years (19–89 years). The mean survival time from surgery was 10.4 ± 7.6 months (0.07–24 months). Of these, 146 (85%) died within 2 years and 5 (3%) were lost to follow-up (all of the five people lived for >6 months after surgery). In total, 121 and 61 patients lived for >6 and >12 months after surgery, respectively (Table 1).

Group 1 consisted of 116 patients with RTS ranging from 0 to 8, which suggests survival time <6 months. Furthermore, group 2 consisted of 42 patients with a score of 9–11, and group 3 consisted of 13 patients with a score of >11. Among the three groups, no age difference was observed, with the mean age of all groups being approximately 55 years. Group 3 had female predominance, whereas group 1 had male predominance. Body measurements were similar between patients of all the three groups, with the mean height and weight being approximately 161 cm and 60 kg, respectively.

Patients in group 1 had a significantly poorer prognosis than patients in the other two groups (Appendix A). Moreover, the 6-month survival rate in group 1 was 42.2%, and the 6-month survival rate was approximately 90% in the other groups. The postoperative 6-month survival probability was mispredicted for 55 (32.2%) patients. Among the 55 patients, survival time of 49 patients (89.1%, all from group 1) was underestimated and that of 6 patients (10.9%) was overestimated. More patients in group 1 had the diagnosis of lung cancer, while no patients in group 3 were diagnosed with lung cancer. In contrast, breast cancer prevalence is the third highest in Taiwan, but no patients in group 1 had breast cancer—significantly different from the remaining groups. The mean survival was 9.1 ± 7.3 (0.39–24) months for patients with lung cancer and 17.0 ± 8.4 (2.14–24) months for patients with breast cancer.

### 3.2. Factors Influencing Survival Time

Forty-nine patients’ 6-month survival (89.1%) was underestimated, and they all belonged to group 1. Therefore, we focused on prognostic factors that influence the postoperative survival at the timepoint to make more accurate survival prediction. Table 2 illustrated that the patients with larger RTS, with the potential of target therapy use, and higher Z-score of BMI (Z-BMI; adjusted for patients’ age and sex) had a better 6-month survival. Of which, the potential of target therapy use had the largest impact, with an odds ratio of 7.08 (95% confidence interval [95% CI], 2.78 to 20.10).

### 3.3. ATS

The potential of target therapy use and Z-BMI were incorporated into RTS to build ATS and to construct a two-step test for better 6-month survival prediction (Figure 2). ATS was initially calculated with undetermined weightings of the two prognostic factors (i.e., *ATS* = *RTS* + *a* × *Target therapy use* + *b* × *Z-BMI*; where use of target therapy =1 and no use =0), and it turned out to illustrate that the corresponding AUCs were larger when a is approximately 1 and b is approximately 4 (Table 3 and Table 4). That is, when a deviates from 1 or b deviates from 4 in both directions, the AUC value will decrease. The discriminatory ability of ATS (0.84) was better than that of RTS (0.76; *p* < 0.001; Figure 3A) in the reference group, and the difference remained significant in the validation group (AUC, 0.76 vs. 0.64; Figure 3B). In the reference group, the recommended cutoff value of ATS was 7.5, with a sensitivity of 0.89 and a specificity of 0.66. Applying the same cutoff value to the validation group yielded a sensitivity of 0.71 and a specificity of 0.83. Patients with an ATS > 7.5 (group A) had a significant better survival than those of ATS < 7.5 (group B; *p* < 0.01; Appendix B).

## 4. Discussion

Whether patients with spinal metastases can survive more than 6 months is a key factor for determining the treatment choice. However, recent studies have suggested that RTS tends to underestimate patients’ survival [11,12,13,14,15]. In this study, we found most mispredictions by RTS came from underestimating the 6-month survival probability of patients in group 1. Therefore, we optimized RTS by incorporating two prognostic factors, namely the potential of target therapy used and Z-BMI. The AUC increased from 0.76 to 0.84 in the reference group and from 0.64 to 0.76 in the validation group. These two prognostic factors are easily retrievable and convenient for clinical use. Physicians can make more personalized medical decisions with the optimization, particularly in predicting whether patients can survive more than 6 months after the surgery. Additionally, more external validations were warranted to test the model’s generalizability and the consumption of this pilot study.

In the era of biologic and checkpoint inhibitors, SORG nomogram and New England Metastatic Spine Score (NESMS) were developed and externally validated for their predictive values [19,20,21]. The former predictive model circumvented the linear limitation, provided more flexible survival estimation, and incorporated the use of systemic therapy as a prognostic factor, while the later incorporated laboratory values to enhance model performance. However, the developers of SORG nomogram mixed the use of chemotherapy, target therapy, and hormone therapy into a single prognostic factor although their curative medication influence on survival time might differ [15,22]. For example, Epidermal growth factor receptor mutation, which is the most common mutation in lung cancer patients in Taiwan (approximately 40% of patients with lung adenocarcinoma) [23,24,25,26], can be treated effectively with target therapy, while the use of chemotherapy might not bring comparably promising survival benefit [23,26,27,28,29,30,31,32]. In this study, we highlighted the importance of target therapy use as a prognostic factor instead of chemotherapy use. If possible, future researcher should analyze the influence of systemic therapy use stratified by their types [33,34,35].

Cancer-associated cachexia is a disorder characterized by loss of body weight with specific losses of skeletal muscle and adipose tissue [36,37,38]. It is driven by several combination of metabolic changes and reduced food intake, including excess catabolism, inflammation, and elevated energy expenditure. It could lead to progressive functional impairment, treatment-related complications, poor quality of life, and cancer-related mortality. In this study, Z-BMI was observed to be a novel prognostic factor for 6-month survival. This might be due to the link between lower BMI level and poorer nutritional status, more severe cachexia, more advanced cancer, and therefore a worsen prognosis. Interestingly, the unadjusted BMI had less influence on 6-month survival partially because the BMI of patients greatly differed by sex.

We have considered taking other nutritional indicators into account, such as albumin level, appetite, or other well-validate evaluation tool [39,40,41,42,43]. However, we failed to include them due to incomplete data and the retrospective nature of this study. For example, only approximately 80% of the patient’s albumin level was recorded. Standardized evaluation tool, such as Glasgow prognostic score or prognostic nutritional index, were not accessible in this study. However, we did not consider this a major limitation since the albumin level is also related to stress and inflammation [44,45,46,47]. Since the spinal surgery could cause huge mental stress, this nature makes serum albumin level an unstable prognostic factor and might lead to its lack of consistency and predictive value. On the other hand, we tried to incorporate the area of psoas muscle in the current prediction model. However, our previous work demonstrated the morphometric analysis might not be an ideal prognostic factor since the incorporation brought insignificant improvement in discriminatory ability [48]. Therefore, we chose Z-BMI as a surrogate in this study due to its higher accessibility.

This study has some limitations. As this was a retrospective study, we could not control many parameters, and they are limited to medical records. Furthermore, we analyzed only the primary tumor site, rather than cancer type. Different cancer types, even in the same organ, might need different treatment and differ in prognosis. Surgery type was not analyzed, and surgeons of surgery were not controlled either. Different surgical techniques might lead to different surgical morbidity and mortality rate. Moreover, all patients in this study underwent surgery, and patients without surgical intervention as a comparative group are lacking. Selection bias exists because patients in a relatively poor condition who could not tolerate surgery were excluded in this study. Although patient’s 12-month survival probability could also guide clinical decision, we failed to improve RTS’s ability to predict survival at the timepoint. However, we did not view it as a major limitation since RTS already performed well in predicting patient’s 12-month survival in this study. Future studies should focus on the limitations of this study.

## 5. Conclusions

We optimized RTS by adding two clinically feasible domains, the potential of target therapy use and the Z-score of BMI, to predict 6-month survival probability in patients with spinal metastases. The incorporation brings much more accurate survival prediction, which could aid spinal surgeons in making more individualized medical decision.

## Figures and Tables

**Figure 1 jcm-11-05391-f001:**
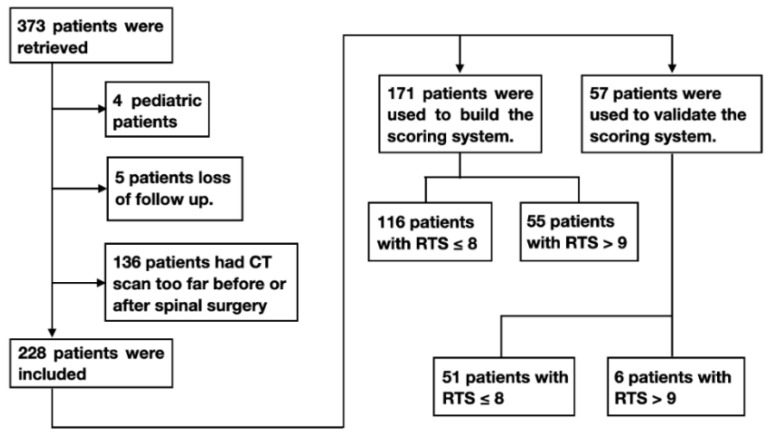
Flow of patient enrollment in this study. Abbreviation: CT = computed tomography; RTS = revised Tokuhashi score.

**Figure 2 jcm-11-05391-f002:**
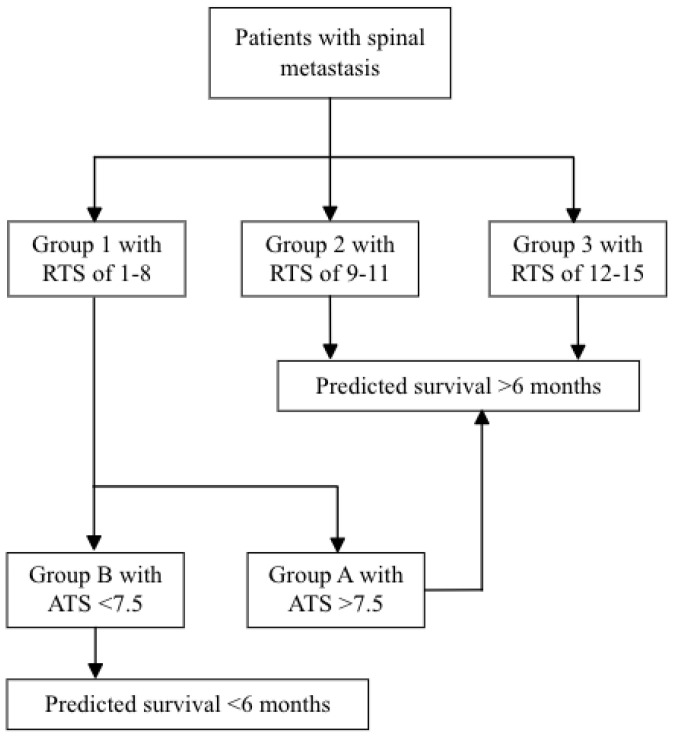
The conceptual of the two-step test. Abbreviation: RTS = revised Tokuhashi score; ATS = adjusted Tokuhashi score.

**Figure 3 jcm-11-05391-f003:**
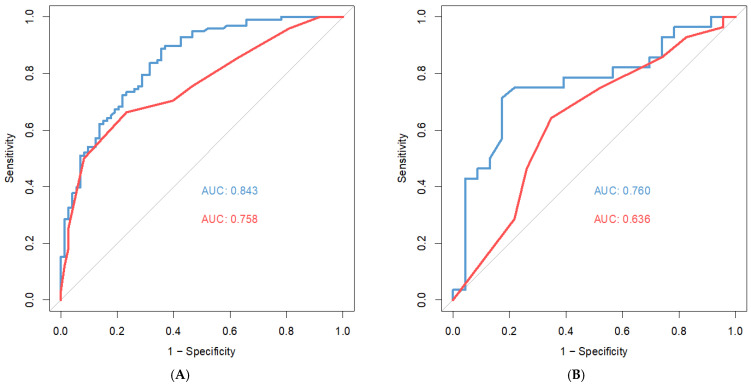
Receiver operating characteristic curves of revised (red) and adjusted (blue) Tokuhashi scores (**A**) in the reference group and (**B**) in the validation group. Abbreviation: AUC = area under curve; ATS = adjusted Tokuhashi score; RTS = revised Tokuhashi score.

**Table 1 jcm-11-05391-t001:** Demographic and clinical characteristics of reference group.

	Group 1 ^a^ (*n* = 116)	Group 2 ^b^ (*n* = 42)	Group 3 ^c^ (*n* = 13)	*p*-Value
**Age**	58.6 (22–89)	57.2 (19–82)	54.5 (31–70)	0.489
19–44	16 (13.8%)	6 (14.3%)	2 (15.4%)	
45–64	65 (56.0%)	25 (59.5)	9 (69.2%)	
≥65	35 (30.2%)	11 (26.2%)	2 (15.4%)	0.847
**Sex (Male/Female)**	77/39	21/21	5/8	0.044
**Body measurements**				
Height	162.3 (141–184)	161.3 (135–175)	161.1 (151–177)	0.772
Weight	60.6 (35–109)	60.0 (36–96)	60.8 (46–78)	0.952
BMI	22.9 (15.8–38.2)	23.1 (15.0–34.8)	23.4 (8.5–28.6)	0.906
**Adjuvant/Neoadjuvant therapy**				
Chemotherapy	91 (78.4%)	33 (78.6%)	13 (100%)	0.001
Target therapy	69 (59.5%)	27 (64.3%)	8 (61.5%)	0.86
Hormone therapy	7 (6%)	9 (21.4%)	10 (76.9%)	<0.001
**Primary origin**				
Lung	56 (48.3%)	8 (19.0%)	0 (0%)	<0.001
Liver	19 (16.4%)	5 (11.9%)	0 (0%)	0.257
Breast	5 (4.3%)	7 (16.7%)	8 (61.5%)	<0.001
Prostate	6 (5.2%)	3 (7.1%)	2 (15.4%)	0.355
Colorectal	5 (4.3%)	1 (2.4%)	0 (0%)	<0.05
Renal	4 (3.4%)	0 (0%)	0 (0%)	<0.05
**Survival**				<0.001
6-month survival rate	42.20%	88.10%	92.30%	<0.001
12-month survival rate	24.10%	52.30%	84.60%	<0.001
**Average RTS**	5.50 (0–8)	9.50 (9–11)	12.30 (12–14)	<0.001

^a^ patients with a revised Tokuhashi score less than 9. ^b^ patients with a revised Tokuhashi score between 9 and 11. ^c^ patients with a revised Tokuhashi score more than 11. Abbreviation: BMI = body mass index; RTS = revised Tokuhashi score.

**Table 2 jcm-11-05391-t002:** Logistic regression for 6-month survival rate in group 1.

Indicators	OR	95% CI	*p*-Value
RTS	1.41	(1.11 to 1.83)	0.005
Chemo	0.67	(0.22 to 1.99)	0.466
Target	7.08	(2.78 to 20.10)	<0.001
Age	1.02	(0.98 to 1.06)	0.276
Sex	0.55	(0.21 to 1.38)	0.220
Z-BMI	1.89	(1.09 to 3.39)	0.027

Abbreviation: RTS = revised Tokuhashi score; Chemo = chemotherapy use; Target = target therapy use; BMI = Body Mass Index; Z-BMI = Z score of BMI of each patient in terms of their age and sex.

**Table 3 jcm-11-05391-t003:** Area under receiver operating characteristic curve of different models.

	*b* = 1	2	3	4	5	6	7	8	9	10
*a* = 1	0.803	0.801	0.788	0.777	0.764	0.749	0.738	0.727	0.719	0.712
2	0.824	0.819	0.808	0.794	0.778	0.761	0.75	0.739	0.729	0.723
3	0.835	0.83	0.816	0/804	0.791	0.773	0.762	0.751	0.741	0.733
4	0.844	0.839	0.825	0.812	0.8	0.786	0.773	0.763	0.752	0.743
5	0.843	0.841	0.83	0.819	0.807	0.792	0.781	0.77	0.759	9.752
6	0.843	0.838	0.831	0.821	0.81	0.799	0.786	0.776	0.767	0.759
7	0.836	0.835	0.828	0.823	0.813	0.801	0.791	0.781	0.772	0.764
8	0.828	0.829	0.825	0.82	0.814	0.804	0.795	0.786	0.776	0.769
9	0.823	0.823	0.821	0.818	0.812	0.805	0.796	0.789	0.78	0.772
10	0.822	0.819	0.817	0.815	0.81	0.803	0.798	0.79	0.784	0.776

“*a*” and “*b*” represent the coefficient in the algorithm of ATS = RTS + *a* × Target therapy use + *b* × Z-BMI. Area under receiver operating characteristic curve is calculated as regards prediction of 6-month survival in group 1. Abbreviation: ATS = adjusted Tokuhashi score; Z-BMI = Z score of BMI of each patient in terms of their age and sex; RTS = revised Tokuhashi score.

**Table 4 jcm-11-05391-t004:** Adjusted Tokuhashi score.

Characteristic	Score
General condition	
Poor (10–40%)	0
Moderate (50–70%)	1
Good (80–100%)	2
Number of extraspinal metastatic foci	
≥3	0
1–2	1
0	2
Number of metastases in vertebral body	
≥3	0
2	1
1	2
Metastases to major internal organ	
Unremovable	0
Removable	1
No metastasis	2
Primary cancer site	
Lung, osteosarcoma, stomach, bladder, esophagus, pancreas	0
Liver, gallbladder, unidentified	1
Others	2
Kidney, uterus	3
Rectum	4
Thyroid, prostate, breast carcinoid tumor	5
Palsy	
Complete (Frankel A, B)	0
Incomplete (Frankel C, D)	1
None (Frankel E)	2
Target therapy	
No use	0
Use	4
Z-BMI	
Total score > 8; Survival > 6 months. Total score < 8; Survival < 6 months.	

Abbreviation: BMI = Body Mass Index; Z-BMI = Z score of BMI of each patient in terms of their age and sex.

## Data Availability

Source data may be shared upon reasonable request to the corresponding author.

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
