# Peer review of "Optimization of Tokuhashi Scoring System to Improve Survival Prediction in Patients with Spinal Metastases"

_jcm, 2022, doi:10.3390/jcm11185391_

Round 1

Reviewer 1 Report (Previous Reviewer 3)

I think that now your work that is suitable for publication; just, in line 158, ( 

ATS=RTS+a×Target therapy use +b×Z-BMI)specify that you decide to put in the equation use of target therapy=1 and no use =0 otherwise it cannot make sense. 

Author Response

Point 1: In line 158, ATS=RTS+a×Target therapy use +b×Z-BMI) specify that you decide to put in the equation use of target therapy=1 and no use =0 otherwise it cannot make sense.

Response 1: Thank you for your asking, and we’re glad to make this right.

Indeed, as you mentioned, if not specifying this, some potential readers might use the euqation wrong. Therefore, we made the following modification accordingly to further clarify the concept.

Line 163:

ATS was initially calculated with undetermined weightings of the two prognostic factors (i.e., ATS=RTS+a×Target therapy use +b×Z-BMI; where use of target therapy=1 and no use =0), and it turned out to illustrate that the corresponding AUCs were larger when a is approximately 1 and b is approximately 4 (Table 3, 4).

Reviewer 2 Report (New Reviewer)

Thank you for submitting your manuscript.  The topic is clearly important however, I think there are some potential area that could strengthen the manuscript presentation and conclusions.

1.  In the introduction, it is not clear if the goal is to create a new score, modify the existing score because of inherent flaws to the score, or add new variables that when added to the RTS create a more predictive model.

2.  Methods:  The methodology is not clearly written and although it is described as a retrospective study it isn't mentioned early.  The rationale for choosing to compare the ATS to the RTS without comparing the RTS to RTS in your patient sample in addition to comparing the RTS to ATS. Since the original RTS was used with the addition of new categories, this may be better considered a pilot. Recommend a statement regarding description of institution, and in table 1 a statement on diversity, and gender differences

Author Response

Point 1: In the introduction, it is not clear if the goal is to create a new score, modify the existing score because of inherent flaws to the score, or add new variables that when added to the RTS create a more predictive model.

Response 2: Thank you for your precious opinion. We would like to state our goal more clearly so the potential readers could have a more distinct figure of this study when reading the introduction.

Therefore, we made the following modification accordingly to further clarify the study goal.

Line 61:

Hence, in this study, we intended to create a more predictive model by adding new variables to the existing RTS in order to distinguish the patients with a survival ≥6 months from those with <6 months [14,15].

Point 2: The methodology is not clearly written and although it is described as a retrospective study it isn't mentioned early.  The rationale for choosing to compare the ATS to the RTS without comparing the RTS to RTS in your patient sample in addition to comparing the RTS to ATS. Since the original RTS was used with the addition of new categories, this may be better considered a pilot. Recommend a statement regarding description of institution, and in table 1 a statement on diversity, and gender differences.

Response 2: Thanks for the valuable reminder and suggestion.

We added the description of the retrospective nature in line 67 so the potential readers might not get confused.

The retrospective research was approved by the institutional review board of the hospital. (approval no.: 202005016RINC).

The rationale of not comparing RTS in historical cohort to the RTS in our patient sample is that several previous studies had already reported the ill performance of RTS in more modern cohorts, as we mentioned in the introduction. Therefore, we felt it was acceptable to make further progress based on the assumption that RTS was outdated due to the fast progress of oncological treatment, without showing the comparison of RTS in historical cohort and in modern cohort. Such compairons could be made, on the other hand, if you believe adding such data could make the article more comprehensive and educative.

As for the suggestion that stating this study as a pilot study, we believe the suggestion was wonderful. We need to tone down our statement since the model indeed needs more further validation support. Therefore, we made the following modification accordingly.

Line 197:

Also, more external validations were warranted to test the model’s generalizability and the consumption of this pilot study.

We added some description of our institution, without leaking the specific infromation due to the blinding policy, in line 68.

We enrolled patients who underwent surgical intervention for pathologically confirmed spinal metastasis, including palliative decompression and aggressive excisional surgery, between January 2012 and December 2017, in a tertiary center in Taiwan.

As for the gender difference, we have already mentioned it in line 127. Hope this meet your standard. As for the diversity, we believe the term is a bit vague. We did not record the patients’ ethnicity, race, or occupation. But according to our Census Bureau, more than 95% of the population was Han Chinese.

This manuscript is a resubmission of an earlier submission. The following is a list of the peer review reports and author responses from that submission.

Round 1

Reviewer 1 Report

Comments to Authors:
The authors presented inaccurate prognosis evaluation by revised Tokuhashi score in their patients and created a new scoring system with a higher accuracy in predicting 6-month survival base on their analyses in this study. The authors mention that target therapy use can prolong patients’ survival and ZZ-BMI is another important influencing factor. I wonder that their adjusted scoring system is better, and that the analyses are appropriate. The analyses and descriptions are complicated and hard to understand.

  1. Figure 1: This figure is reverse left and right, upside down. It is hard to see and understand it.

  1. Patients included in this study: The study cohort was biased because All 373 patients enrolled in the study and 228 included in the study underwent surgery. In the introduction, the authors mentioned that patients with scores of ≤8 are predicted to survive <6 months, nonsurgical treatments are suggested. However, 116 patients (more than half of the cases in the analysis!!) in Group 1 with scores of ≤8 underwent surgery. Therefore, I think that the cohort was biased and inappropriate, and that the analyses were not fair. Especially, the patients with scores of ≤8 treated with conservative treatments should be included in the analyses.

  1. Survival time (Table 1): The patients were followed for a maximum of 2 years after the index surgery. That means some patients survived for more than 2 years. In Table 1, survival times in Groups 1-3 were presented as mean. The longest time was 2 years but it was incorrect. Similarly, the mean survival times were also incorrect. In my opinion, the median survival time should be presented and analyzed.

  1. ZZ-BMI: The authors mentioned that ZZ-BMI was an important factor associated with prognosis. It is included in their adjusted Tokuhashi score. However, a more detailed explanation of ZZ-BMI is required. In this kind of universal scoring systems including Tokuhashi score, the included factors should be easily available in the clinical setting. I do not think that Z-BMI and ZZ-BMI are suitable and easy-available factors.

  1. I understand that an accurate prediction of survival <6 months is important for surgical indication. However, for the accuracy and validity of their scoring system based on RTS, a prediction of survival <12 months is also important. I cannot understand that it is not considered as a better one without an analysis of prediction of survival <12 months.

  1. Table A3: The authors mentioned that patients in group A had a high BMI and weight, good prognosis, and high 6-month survival rates, and that most patients in group A had received target therapy. This result is not surprising, since the author classified them using the scoring that included that factor.

  1. As mentioned by the authors in the discussion, low BMI does not directly indicated malnutrition. Glasgow prognostic score, prognostic nutritional index, neutrophil-lymphocyte ratio, and platelet-lymphocyte rate are famous in the nutrition assessment in cancer patients. The authors should consider an analyses using other nutrition assessments when they concluded that nutrition status was important for prognosis in their study.

Reviewer 2 Report

The authors performed a retrospective review a large cohort of operated metastatic spine-tumor patients to establish a new adjusted Tokuhashi score and validated it based on a sequential cohort of patients. The adjusted Tokuhashi score adds ZZ-BMI and the use of targeted therapy to the revised targeted Tokuhashi score. Adding these two parameters improved the less than six-month survival prediction from 57.8% to 75.9% in the newer scoring system. 

In modern spine oncology, the concept of using survival as the sole determinant of treatment limits the utility of even the adjusted Tokuhashi score. Survival is a very important consideration in treatment, but the designation of those with extended survival requiring en bloc or gross total excisional surgery vs those with limited (ie less than 6 month survival) requiring posterior fusion +/- laminectomy does not really hold up in contemporary treatment paradigms such as NOMS. The addition of stereotactic radiosurgery and minimally invasive techniques to stabilize the spine has made the need for excisional procedures less important for metastatic tumors. Survival remains a significant determinant of choosing a treatment option, but the type of treatment is no longer determined by expected survival. This paper would be better focused on simply presenting a predictive model of survival rather than using survival to determine treatment.

ZZ-BMI is not a designated term in the literature and is not explained well in the test. It is unclear what the authors mean by this designation. Standard BMI reporting would be more impactful.

Figure 1 is inverted and should be corrected. 

The SORG nomogram and New England Metastatic Spine Score are both validated predictive survival models developed in the era of biologic and checkpoint inhibitors. SORG nomogram has been externally validated. These two scoring systems should be acknowledged and compared in the discussion. 

Reviewer 3 Report

Dear Authors, thanks for letting me review this article. 

I'm highly interested in the topic and I think that you write a potentially very good paper. I think that could be considered for publications but I have some big concerns: 

  • The style is not so fluent and the article is quite difficult to read and understand. There is a potential contradiction inside the article. Y0u stated in line 111-114

    "We divided the 228 patients into two groups. In total, 171 patients who underwent surgery before December 31, 2016, were examined to evaluate the accuracy of RTS and  determine factors that affect prognosis. The remaining 57 patients who underwent spinal surgery after January 1, 2017, were examined to validate the reliability of a new score, that is, adjusted Tokuhashi score" 

    But in reality you first analyzed the first group, then you use the data for building ATS then re-analyze the first group splitting into subgroups 1A and 1B then you finally analyze validation group.  In material and methods, you should explain the process in order to make the article more fluid.  
  • There is a potential bug that can jeopardize all the study: in line 255-258, you state 

    "Of the 57 patients in the validation group, 51 patients with RTS ≤8 had a 6-month survival rate of 45.1%. ATSs of 25 patients were <8 and those of 26 patients were ≥8, and the 6-month survival rates were 28% and 80.7%, respectively. The total accuracy rate of  predicting 6-month survival by using ATS was 76.5%, which is also much higher than that obtained by using RTS (54.9%)"

    In this statement, you do not demonstrate anything more than that scoring 8 is probably the problem. Although you show  in fig 2 a bigger AUC for ATS versus RTS, you should demonstrate that in  validation group there is a difference applying the ATS VS RTS with the different cutoff point ( as you stated inline 179-181)
  • I'm not so good in statistics, so I rather prefer a legend for the two columns in table 2 , 3 and 4 . What difference in regressions they show? ( apart from the difference in Z-BMI and ZZ-BMI)
  • can you show a more detailed origin of tumors? because the " other" ( not lung-prostate-liver-breast) group accounts for 25%  in group 1, 23% in group 3 and an astonishing 45% in group 2..does not seems possible to avoid description. i.e., colorectal  and Kidney cancer should be included.
  • There is a  surgically-related consideration to do:  sometimes ( as in clear cell carcinoma)  target therapy is administered after a PD, and often the progression is a spine met. Do you think  that in scoring system we must consider target therap as a " potential use of target therapy"  or just " we already use target therapy"? can you state this?  
  • Probably is due to document loading but figure  1 is flipped